# Josephson effect in strongly disordered metallic wires

**Mustafa E. Ismagambetov[1], Aleksey V. Lunkin[2] and Pavel M. Ostrovsky[1⋆]**

**1** Max Planck Institute for Solid State Research, 70569 Stuttgart, Germany
**2** Nanocenter CENN, Jamova 39, SI-1000 Ljubljana, Slovenia

⋆ p.ostrovsky@fkf.mpg.de

## Abstract

We study localization effects in Josephson junctions with two superconductors connected by a strongly disordered metallic wire of length $L$. The conventional description of the Josephson effect in such systems, based on the quasiclassical Usadel equation, neglects electron interference and is only applicable when $L$ is shorter than the localization length $\xi$ in the wire. We develop a more general theory for the Josephson effect using the nonlinear sigma model that fully accounts for electron interference, and hence localization. We show that for $L \gg \xi$, three qualitatively different regimes of the Josephson current arise depending on the ratio of the superconducting order parameter $\Delta$ and the mean level spacing in the localization volume $\Delta_\xi$. We derive the average supercurrent as a function of the phase difference for all three regimes. Quite unexpectedly, we observe that the Ambegaokar-Baratoff relation between the average critical current and the normal-state conductance still holds in the strongly localized state when $\Delta_\xi \gg \Delta$ and $\xi \ll L \ll (\xi/\pi^2) \ln^2(\Delta_\xi/\Delta)$.

# 1 Introduction

Transmission of Cooper pairs through insulating barriers is known since the discovery of the Josepshon effect [1] in superconductor-insulator-superconductor (SIS) structures. The maximum available value of the supercurrent in such junctions (critical current) is proportional to the normal state conductance of the barrier, a statement known as the Ambegaokar-Baratoff relation [2]. When the barrier between two superconductors is realized by a band insulator, which is typical for SIS junctions, both the critical current and normal conductance decay exponentially on the atomic scale with the thickness of the insulating layer.

Another realization of the Josephson effect is provided by SNS junctions with a normal metal between two superconducting leads. Cooper pairs propagate much easier in the normal metal that provides a relatively high density of states near the Fermi energy. The standard theoretical description of such a Josephson effect is in terms of Andreev reflection at the NS boundaries [3] and the ensuing superconducting correlations in the normal part of the junction (proximity effect). When electron transport in the metal is diffusive due to elastic scattering on impurities, two qualitatively different regimes of long and short SNS junction are distinguished. For a relatively short junction whose thickness is small compared to the superconducting coherence length, the Ambegaokar-Baratoff relation still holds, and the critical current decays inversely proportional to the thickness of the normal layer according to Ohm's law [4]. In longer junctions exceeding the superconducting coherence length the critical current is proportional to the Thouless energy of the junction $E_{\mathrm{Th}} = D/L^2$ (here $D$ is the diffusion constant) and hence decays as $L^{-2}$ [5,6]. For such long junctions, the Ambegaokar-Baratoff relation is manifestly violated.

Classical results for the Josephson current mentioned above completely disregard possible quantum interference effects in the normal part of the SNS junction. Such effects are well known in the theory of quantum transport and lie at the core of Anderson localization [7]. Localization is a very rich phenomenon that arises most prominently in low-dimensional samples at low temperatures. It has numerous manifestations ranging from weak localization and magnetoresistance [8] in metallic films and wires to the physics of topological insulators and superconductors [9]. An important characteristic feature of localization is its universality. Quantum interference of electrons in disordered media on long scales is insensitive to the microscopic details of the material and depends only on the general symmetries of the system. Although localization phenomena in general are quite well understood, the interplay between localization and superconducting proximity effect is much less studied. The purpose of this paper is to fill this gap.

We will consider SNS junctions with a narrow disordered metallic wire connecting two superconductors. In such a geometry, localization effects play a significant role. They are characterized by a typical length scale $\xi$—localization length. Individual electron wave functions in the disordered metallic wire are localized on this scale making the sample insulating once its length exceeds $\xi$. Classical theory of the Josephson effect discussed above breaks down in this limit. Our goal is to develop a more general quantum description valid for SNS junction of arbitrary length both shorter and longer than $\xi$.

Josephson current in strongly localized samples was studied experimentally by Frydman and Ovadyahu [10]. They measured critical currents of wide SNS sandwiches with two Pb superconductors separated by a thin film of disordered $\mathrm{InO}_x$. The thickness of the metallic

layer varied from sample to sample in the interval 70–1000Å while the localization length was estimated to be 40Å. Surprisingly, the critical current in such junctions was still as high as predicted by the Ambegaokar-Baratoff relation whenever the thickness exceeded 400Å, well in the localized regime. At the same time, thinner samples exhibited a critical current one order of magnitude less. This counterintuitive observation suggests that the naïve picture of localization simply suppressing the Josephson current when the length exceeds $\xi$ is not fully correct.

In this paper, we develop a fully quantum theory of the Josephson effect based on the nonlinear sigma model [11]. This is an effective field theory describing electron transport in disordered metals taking into account all quantum interference effects. It is rather straightforward to include superconductivity in the sigma-model formalism [12]. Classical results for the Josephson current (neglecting localization) can be extracted from the sigma-model description by applying quasiclassical approximation to the action. Finding the optimal configuration of the sigma-model field that minimizes the action is fully equivalent to solving the Usadel equation [13] for the quasiclassical Green function. This way we can reproduce the classical results directly from the sigma model. When the length of the junction exceeds $\xi$, the quasiclassical approximation is no longer valid and a more accurate treatment of the sigma model beyond saddle point approximation is required. We perform this calculation in the present paper and identify three qualitatively different regimes of the Josephson current in the strongly localized metallic wires depending on the ratio of the superconducting order parameter $\Delta$ and the mean level spacing in the localization volume $\Delta_\xi = D/\xi^2$.

The structure of the paper is as follows. In Sec. 2, we briefly review the quasiclassical theory of the Josephson effect based on the Usadel equation and derive the results for the short and long junction limits. In Sec. 3, we formulate a fully quantum approach to the problem based on the nonlinear sigma model. The transfer matrix method for solving the non-linear sigma model is outlined in Sec. 4. Explicit calculation of the average Josephson current in the localized limit is performed in Sec. 5. We identify three qualitatively different transport regimes and find the corresponding current-phase relations. The results are summarized and discussed in Sec. 6. Some technically involved details of the calculation are given in Appendix A.

## 2   Classical SNS junction

We consider an SNS junction made of a normal disordered metallic wire of length $L$ connecting two bulk superconducting leads. Both leads have the same superconducting gap $\Delta$ and we assume ideal contacts between the normal part and each of the superconductors. A schematic representation of the junction is shown in Fig. 1. Our main goal is to study the impact of Anderson localization in the wire on the Josephson effect. We will distinguish two qualitatively different situations when the length of the junction $L$ is either shorter or longer than the localization length

$$\xi = \pi \nu D. \tag{1}$$

Here and throughout, $\nu$ is the electron density of states per one spin component times the wire cross-section and $D$ is the diffusion constant. While the localization effects are strongest in the case of a long wire, in this Section we review the results in the classical limit $L \ll \xi$.

### 2.1   Usadel equation

Disorder in the normal part of the junction implies diffusive electron propagation. It can be described in terms of the quasiclassical Green functions by the Usadel equation [13]. The

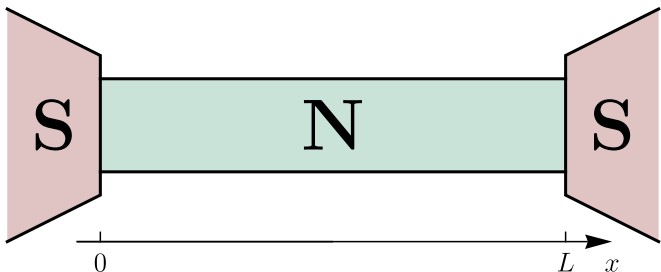

Figure 1: Schematic representation of a long SNS junction. The normal part of the junction is a strongly disordered metallic wire of the length $L$.

Usadel equation is formulated for a $2 \times 2$ matrix $Q$ that satisfies the condition $Q^2 = 1$. This matrix represents the quasiclassical Green function in the Nambu-Gor'kov space. The Usadel equation can be thought of as the Euler-Lagrange equation for the following action:

$$S = \frac{\pi \nu}{4} \int dx \, \mathrm{tr} \left[ D \left( \frac{\partial Q}{\partial x} \right)^2 - 4 \begin{pmatrix} \epsilon & \Delta \\ \Delta^* & -\epsilon \end{pmatrix} Q \right]. \tag{2}$$

Here $\epsilon$ is a positive Matsubara energy, and $\Delta$ is the complex superconducting order parameter. The latter is present only inside superconducting leads.

In the normal state, when both leads have $\Delta = 0$, conductance of the junction is given by the standard Drude formula

$$G = 2e^2 \nu D / L. \tag{3}$$

This expression is valid for a wire shorter than the localization length (1).

Usadel equation minimizes the action (2). It can be derived by varying the action under the condition $Q^2 = 1$. Inside the normal part of the junction, where $\Delta = 0$, the Usadel equation reads

$$D \frac{\partial}{\partial x} \left( Q \frac{\partial Q}{\partial x} \right) = \epsilon [\tau_z, Q]. \tag{4}$$

We use the notation $\tau_{x,y,z}$ for Pauli matrices in the Nambu-Gor'kov space.

Usadel equation (4) has the form of a nonlinear diffusion equation. It provides quasiclassical description of the system neglecting possible interference phenomena. It should be emphasized that, while the action (2) does provide an accurate saddle-point description of the system, it cannot be canonically quantized in order to take into account localization effects. This can be achieved only within an extended approach of the non-linear sigma model as will be explained in the next Section.

The nonlinear condition $Q^2 = 1$ can be explicitly resolved by the following parametrization:

$$Q = \tau_z \cos \theta + \sin \theta (\tau_x \cos \phi + \tau_y \sin \phi). \tag{5}$$

Parameters $\theta$ and $\phi$ can be thought of as spherical coordinates on a two-dimensional unit sphere. Then the Usadel equation (4) is exactly a diffusion equation on such a sphere with an external force proportional to $\epsilon$.

Boundary conditions can be found by minimizing the action (2) inside superconducting leads. Neglecting the gradient term there, we have

$$\theta(0) = \theta(L) = \theta_S = \arctan \frac{\Delta}{\epsilon}, \qquad \phi(L) - \phi(0) = \phi. \tag{6}$$

Here $\phi$ is the phase difference between the two superconductors; also from here and for the rest of the paper $\Delta$ is the absolute value of the order parameter, which is the same in both leads.

Solution to the Usadel equation provides the value $S_{\text{min}}$ of the minimized action (2) as a function of $\phi$ and $\epsilon$. The Josephson current can be then found as the spectral integral

$$I(\phi) = \frac{2e}{\hbar} \int_0^\infty \frac{d\epsilon}{\pi} \frac{\partial S_{\text{min}}}{\partial \phi}. \qquad (7)$$

The Usadel action (2) involves two characteristic energy scales: superconducting gap $\Delta$ and the Thouless energy $E_{\text{Th}} = D/L^2$. The ratio of these scales determines two distinct limiting regimes: classical short junction ($E_{\text{Th}} \gg \Delta$) and classical long junction ($E_{\text{Th}} \ll \Delta$).

## 2.2 Short SNS junction

Let us first consider the classical short junction limit. In this case we can neglect $\epsilon$ in the Usadel action (2). The action then has full rotational invariance in the normal part of the junction. A symmetry transformation

$$Q \mapsto \tilde{Q} = U^{-1} Q U, \qquad (8)$$

where $U$ is any unitary matrix, preserves the form of the action. This transformation is equivalent to the rotation of the two-dimensional sphere parametrized by the angles $\theta$ and $\phi$ as in Eq. (5). This rotational symmetry implies that the minimized action $S_{\text{min}}$ depends only on the arc distance between the initial and final points $Q(0)$ and $Q(L)$. To simplify further calculations we will apply a rotation (8) that brings the boundary conditions (6) to the following form:

$$\tilde{Q}(0) = \tau_z, \qquad \tilde{Q}(L) = \tau_z \cos\tilde{\theta} + \tau_x \sin\tilde{\theta}. \qquad (9)$$

The angle $\tilde{\theta}$ is exactly the arc distance between two boundary values of $Q$:

$$\cos\tilde{\theta} = \cos^2\theta_S + \sin^2\theta_S \cos\phi = \frac{\epsilon^2 + \Delta^2 \cos\phi}{\epsilon^2 + \Delta^2}. \qquad (10)$$

Optimal function $\tilde{Q}(x)$ that solves the Usadel equation (4) connects the two points (9) by an arc of a great circle on the sphere: $\tilde{\theta}(x) = (x/L)\tilde{\theta}$ and $\tilde{\phi}(x) = 0$. We can readily calculate the minimized action (2) for this solution and then find the supercurrent using Eq. (7),

$$I(\phi) = \frac{\pi G \Delta}{e} \operatorname{arctanh}\left(\sin\frac{\phi}{2}\right) \cos\frac{\phi}{2}. \qquad (11)$$

This current-phase relation (11) is shown in Fig. 2 with the blue line. It was first derived in Ref. [4] using the Usadel equation as described above. An alternative derivation of the same result was proposed in Ref. [14] where the Josephson current was expressed via transmission probabilities of the junction in the normal state. The relation (11) can then be obtained by using the Dorokhov distribution of transmission probabilities [15] for a diffusive metallic wire.

We have expressed the result (11) in terms of the Drude value (3) of the normal conductance of the junction. We see that the supercurrent is of the order of $G\Delta/e$ exactly as suggested by the Ambegaokar-Baratoff relation [2] for the usual tunnel Josephson junction (with $G = 1/R_N$ being the inverse resistance in the normal state of the system). Hence we conclude that the Ambegaokar-Baratoff relation holds for the classical short SNS junctions.

## 2.3 Long SNS junction

A classical long SNS junction is characterized by the inequality $E_{\text{Th}} \ll \Delta$. In this limit, we should retain the energy term in the action (2). The integral (7) converges at $0 < \epsilon \lesssim E_{\text{Th}}$, hence we can set $\theta_S = \pi/2$ in the boundary conditions (6). This effectively removes the energy scale $\Delta$ from our analysis. If we measure the coordinate $x$ in units of $L$ and energy $\epsilon$ in units

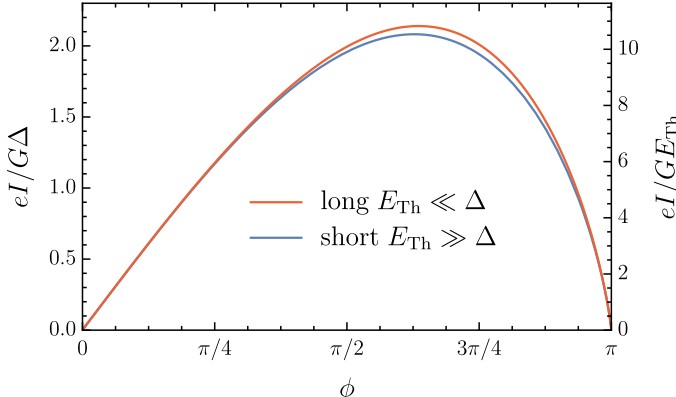

Figure 2: Current-phase relations in a classical SNS junction. Blue curve depicts the supercurrent in the short junction limit as given by Eq. (11) with the corresponding scale on the left side of the frame. Red curve shows the supercurrent in the long junction regime (12) with the scale on the right side. The curves are scaled to have the same slope at $\phi = 0$ in order to demonstrate their numerical similarity.

of $E_{\text{Th}}$, it will render the Usadel equation (4) fully dimensionless with the only parameter being the phase difference $\phi$ incorporated in the boundary conditions. This dimensionless Usadel equation can be solved numerically and then the spectral integral (7) determines the supercurrent

$$I(\phi) = \frac{GE_{\text{Th}}}{e} f(\phi). \tag{12}$$

The function $f(\phi)$ is shown in Fig. 2 with the red line. It has a maximum $f \approx 10.8$ at the phase difference $\phi \approx 2.00$. These values were first found in Ref. [16] and the full current-phase relation was presented in Ref. [6].

A long SNS junction apparently violates the Ambegaokar-Baratoff relation since the critical current is proportional to $E_{\text{Th}}$ instead of $\Delta$. At the same time, functional dependence of the Josephson current on the phase difference in the limits of short and long junction turns out to be very similar as can be seen in Fig. 2. While the shapes of the two functions differ numerically by less than 2%, the current itself has parametrically different prefactors in the two cases.

## 3 Sigma-model description

When the length of the normal wire exceeds localization length, $L \gg \xi$, interference effects become strong and the simple semiclassical approach based on the Usadel equation is no longer valid. A more general formalism that accurately takes into account localization phenomena is required instead. Such formalism is the nonlinear supersymmetric sigma model [11]. This is an effective field theory formulated in terms of the supermatrix $Q$ that generalizes the quasiclassical matrix Green function used in the Usadel equation. The action of the nonlinear sigma model looks quite similar to the quasiclassical action (2):

$$S[Q] = -\frac{\pi \nu}{8} \int_0^L dx \, \text{str} \left[ D \left( \frac{\partial Q}{\partial x} \right)^2 - 4\epsilon \Lambda Q \right]. \tag{13}$$

Here the new extended matrix $Q$ has the size $8 \times 8$ and operates not only in the Nambu-Gor'kov space but also in the additional particle-hole space and in the superspace of bosons and fermions. Particle-hole structure arises due to time-reversal symmetry of the problem; we

will denote Pauli matrices in this space as $\sigma_{x,y,z}$. Superstructure of $Q$ is introduced to fix the value of the total partition function equal to 1 and facilitate disorder averaging of the free energy.

Similar to the quasiclassical Green function, the extended matrix $Q$ obeys the nonlinear constraint $Q^2 = 1$ and can be represented as the rotated matrix $\Lambda$:

$$Q = T^{-1} \Lambda T, \qquad \Lambda = \sigma_z \tau_z. \tag{14}$$

This means that $Q$ belongs to some multidimensional analogue of a unit sphere—symmetric space—and $\Lambda$ is a special point in this space. In addition, $Q$ obeys a linear constraint, which follows from the time-reversal symmetry of the system,

$$Q = \bar{Q} = C^T Q^T C, \qquad C = \sigma_x \begin{pmatrix} \tau_x & 0 \\ 0 & i\tau_y \end{pmatrix}_{\text{BF}}. \tag{15}$$

Conditions (14) and (15) define the supermanifold of the nonlinear sigma model of the orthogonal symmetry class. The structure of this manifold is universal; it always arises in systems with preserved time-reversal and spin symmetries. There are in total 8 real and 8 Grassmann coordinates on this manifold. If we disregard Grassmann variables, the manifold of $Q$ splits into a product of bosonic and fermionic sectors with the structure $O(2,2)/O(2) \times O(2)$ and $\text{Sp}(4)/\text{Sp}(2) \times \text{Sp}(2)$, respectively. Bosonic part of this manifold is equivalent to a product of two two-dimensional hyperboloids while the fermionic sector represents a single four-dimensional sphere. For convergence of integrals over $Q$ it is important that the bosonic/fermionic sectors are noncompact/compact.

We introduce explicit coordinates on the sigma-model manifold by using the Cartan decomposition of the matrix $T$ in Eq. (14). Following Efetov [11], we define

$$Q = U^{-1} \Lambda e^{\hat{\theta}} U, \qquad \hat{\theta} = \begin{pmatrix} \theta_{\text{B}} \sigma_y \tau_z + \theta_{\text{B2}} \sigma_z \tau_y & 0 \\ 0 & i\theta_{\text{F}} \sigma_y \tau_z \end{pmatrix}_{\text{BF}}. \tag{16}$$

Here the matrix $\hat{\theta}$ is chosen such that $\{\hat{\theta}, \Lambda\} = 0$ and $\bar{\hat{\theta}} = -\hat{\theta}$; in addition, three generators of $\hat{\theta}$ commute with each other. The matrix $U$ contains all other parameters of the manifold and obeys $U^{-1} = \bar{U}$ and $[U, \Lambda] = 0$. Parameters $\theta_{\text{B}}$ and $\theta_{\text{B2}}$ are just the polar angles on the two hyperboloids in the bosonic sector; they take any positive values $\theta_{\text{B,B2}} > 0$. The angle $\theta_{\text{F}}$ is the polar angle on the four-dimensional sphere in the fermionic sector; it belongs to the interval $0 < \theta_{\text{F}} < \pi$. In the semiclassical limit, the angle $\theta_{\text{F}}$ is identical to the angle $\theta$ of the Usadel equation, cf. Eq. (5). The same is true for the bosonic sector after analytic continuation $\theta_{\text{B}} \mapsto i\theta$ and setting $\theta_{\text{B2}} = 0$.

We also introduce analogues of the superconducting phase [parameter $\phi$ in Eq. (5)] both in the bosonic and fermionic sectors:

$$U = e^{i\hat{\phi}/2} V, \qquad \hat{\phi} = \sigma_z \begin{pmatrix} \phi_{\text{B}} & 0 \\ 0 & \phi_{\text{F}} \end{pmatrix}_{\text{BF}}. \tag{17}$$

These parameters vary in the interval $0 < \phi_{\text{B,F}} < 2\pi$. Remaining coordinates on the sigma-model manifold are contained in the matrix $V$; explicit parametrization of this matrix will not be important for our calculation.

The action (13) of the sigma model can be used to calculate any average observable quantities of the disordered system. For our problem of the Josephson effect we will consider two such quantities: average conductance $\langle G \rangle$ in the normal state and average supercurrent $\langle I(\phi) \rangle$ as a function of phase difference in the superconducting state of the system. Both quantities can be related to the evolution operator of the sigma model

$$W(Q_f, Q_i, L) = \int_{Q(0)=Q_i}^{Q(L)=Q_f} DQ \, e^{-S[Q]}. \tag{18}$$

This evolution operator has the form of a path integral on the sigma-model manifold. It sums over trajectories $Q(x)$ with the fixed initial and final points $Q_i$ and $Q_f$ while the real space coordinate $x$ plays the role of a fictitious time variable.

To find the normal conductance of the system, we fix the initial point $Q_i = \Lambda$ and the energy $\epsilon = 0$. Under these assumptions, the evolution operator depends only on the values of $\hat{\theta}$ in the final point of trajectory $Q_f$. Average conductance is then expressed as

$$\langle G \rangle = -\frac{e^2}{\pi\hbar} \left[ \frac{\partial^2 W}{\partial \theta_{\rm B}^2} + \frac{\partial^2 W}{\partial \theta_{\rm F}^2} \right]_{\substack{\hat{\theta}=0 \\ \epsilon=0}}. \tag{19}$$

For a relatively short junction $L \ll \xi$ we can replace the path integral (18) by the contribution of a single classical trajectory that minimizes the action. This is exactly the trajectory that obeys the Usadel equation with the given boundary conditions. Then Eq. (19) reproduces the Drude result (3).

In order to calculate supercurrent, we should apply different boundary conditions. On both sides of the junction we fix the value of angles $\hat{\theta}$ such that $\theta_{\rm B} = i\theta_S$, $\theta_{\rm F} = \theta_S$, $\theta_{\rm B2} = 0$, cf. Eq. (6). In addition, we set zero phases $\phi_{\rm B,F} = 0$ on the left end of the junction at $x = 0$ and consider the evolution operator (18) as a function of the two phases on the right end at $x = L$. The current is then given by the spectral integral

$$\langle I(\phi) \rangle = \frac{2e}{\hbar} \int_0^\infty \frac{d\epsilon}{\pi} \frac{\partial W}{\partial \phi_{\rm B}} \bigg|_{\phi_{\rm B,F}=\phi}. \tag{20}$$

This is a generalization of Eq. (7) used in the quasiclassical calculation of the previous Section. After taking derivative with respect to $\phi_{\rm B}$ we should set both phases on the right end of the junctions equal to $\phi$—superconducting phase of the right lead. For a relatively short junction $L \ll \xi$, we can replace the full path integral (18) by the contribution of a single trajectory that minimizes the action (13). This will reproduce the classical results (11) and (12).

Now we have all the necessary ingredients to calculate the supercurrent in long junctions taking into account localization effects. It will amount to calculating the path integral (18) beyond quasiclassical approximation.

## 4 Evolution equation

Full path integral (18) represents the quantum evolution amplitude on the sigma-model manifold between points $Q_i$ and $Q_f$ with the action (13) in fictitious time $L$. It is similar to the standard Feynman path integral in usual quantum mechanics. Hence we can derive an equivalent differential equation [11] for $W$ similar to the Schrödinger equation. Sigma-model manifold will play the role of the real space with $Q$ being the coordinate and $L$ the imaginary time. Evolution equation has the form

$$-\xi \frac{\partial W}{\partial L} = HW, \qquad H = -\frac{\nabla_Q^2}{2} + \frac{\epsilon}{2\Delta_\xi} \operatorname{str}(\Lambda Q), \qquad W(Q_f, Q_i, L=0) = \delta(Q_f - Q_i). \tag{21}$$

The operator in the right-hand side is called the transfer-matrix Hamiltonian. It is analogous to the Hamiltonian operator in quantum mechanics and can be derived from the action (13) in exactly the same way as the usual Hamiltonian of quantum mechanics is derived from the classical action. We will refer to Eq. (21) as the evolution equation.

Hamiltonian (21) is the sum of two terms similar to the kinetic and potential energy. Kinetic energy is written in terms of $\nabla_Q^2$—the Laplace-Beltrami operator on the sigma-model manifold.

Explicit form of this operator is given in Appendix A. Potential energy is just a function of $Q$ and is proportional to the Matsubara energy $\epsilon$ measured in units of the mean level spacing in the localization volume

$$\Delta_\xi = \frac{D}{\xi^2}. \tag{22}$$

Let us first discuss the calculation of average conductance in a long disordered wire with the help of Eq. (19). From the very beginning we can set $\epsilon = 0$ so that our Hamiltonian will contain only the kinetic term. Eigenfunctions of the Laplace-Beltrami operator, and hence of the Hamiltonian, are similar to the standard set of spherical functions on a usual sphere. Moreover, since the initial point in Eq. (19) is $Q_i = \Lambda$, the evolution operator $W$ contains only the eigenfunctions that are invariant under rotations $Q_f \mapsto U^{-1} Q_f U$ with any matrix $U$ that commutes with $\Lambda$. Such eigenfunctions are called zonal spherical functions. On a usual sphere, they correspond to the functions that have zero projection of the angular momentum on the $z$ axis and hence are independent of the azymuthal angle. For a general symmetric space, zonal spherical functions can be explicitly constructed using the Iwasawa decomposition of the matrix $T$ in Eq. (14). For supersymmetric manifolds such a construction was first proposed in Refs. [17, 18] and later carried out explicitly in Ref. [19]. Eigenfunctions relevant for our problem are given in Appendix A.

For a relatively long wire $L \gg \xi$, only the lowest eigenstates of the Hamiltonian in Eq. (21) are relevant. These states are labeled by a single real positive number $q$, see Eq. (A.5). We expand the evolution operator (18) in the eigenfunctions and set $\theta_{B2} = 0$ as required by Eq. (19). The resulting expression for $W$ is

$$W(\theta_B, \theta_F) = 1 - \frac{\pi}{2} \int_0^\infty dq \, \tanh^2(\pi q)(\cosh \theta_B - \cos \theta_F)$$
$$\times P_{-1/2+iq}(\cosh \theta_B) \exp[-(q^2 + 1/4)L/\xi] + \ldots \tag{23}$$

Omitted terms contain higher eigenfunctions and decay faster in the limit $L \gg \xi$.

Average conductance can be now readily found from Eq. (19). In the limit $L \gg \xi$ only small values of $q$ contribute to the integral in Eq. (23). Expanding the pre-exponential factor in small $q$ and using Eq. (19), we obtain the result

$$\langle G \rangle = \frac{\pi e^2}{4\hbar} \left( \frac{\pi \xi}{L} \right)^{3/2} e^{-L/4\xi}. \tag{24}$$

This value of conductance was first obtained in Ref. [17]. Let us also point out that in the localized limit, conductance is a strongly fluctuating quantity. Its distribution is roughly log-normal with the average value given by Eq. (24) but the typical value decaying much faster as $G_{typ} \propto e^{-L/\xi}$.

## 5 Average supercurrent in the localized limit

In this Section we will apply the formalism developed above to the calculation of the super-current in the localized limit $L \gg \xi$. In the most of this Section we will assume $\Delta_\xi \gg \Delta$ (this assumption will be removed in the very end) and gradually increase the length of the junction $L$ beyond the localization length $\xi$. Unlike normal conductance, calculation of the supercurrent involves nonzero Matsubara energies, see Eq. (20), and hence requires the knowledge of the Hamiltonian (21) eigenfunctions in the presence of a finite potential term. The integral over Matsubara energies in Eq. (20) converges at $\epsilon \lesssim \Delta$ due to boundary conditions of the evolution operator. Hence the prefactor of the potential term in the Hamiltonian is small for

all relevant values of $\epsilon$. This fact was already used to neglect $\epsilon$ term in the Usadel equation (4) and obtain the supercurrent (11) in the classical short junction. Now with $L \gg \xi$ we will still neglect the potential term in the Hamiltonian and later establish validity limits of this approximation.

In the absence of potential term the Hamiltonian, given by the Laplace-Beltrami operator, is fully symmetric under any rotations on the sigma-model manifold. This symmetry was already used previously in the classical limit and allowed us to rotate the initial point for the Usadel equation to $\Lambda$. We will apply the same trick to the sigma model and introduce the rotated matrix

$$\tilde{Q} = e^{i\theta_S \sigma_y \tau_z/2} Q e^{-i\theta_S \sigma_y \tau_z/2}. \tag{25}$$

This will indeed bring the initial point of the evolution operator to $\tilde{Q}_i = \Lambda$. The final point $\tilde{Q}_f$ will contain the angles $\tilde{\theta}_{\text{B,F}}$ from Eq. (10):

$$\cosh \tilde{\theta}_{\text{B}} = \cos^2 \theta_S + \sin^2 \theta_S \cos \phi_{\text{B}}, \tag{26a}$$

$$\cos \tilde{\theta}_{\text{F}} = \cos^2 \theta_S + \sin^2 \theta_S \cos \phi_{\text{F}}, \tag{26b}$$

$$\tilde{\theta}_{\text{B2}} = 0. \tag{26c}$$

We can directly substitute the evolution operator (23) with these rotated angles into Eq. (20). The integral over $q$ is determined by the small values $q \lesssim \sqrt{\xi/L} \ll 1$ and has the same form as for the average conductance. The remaining energy integral provides the result

$$\langle I(\phi) \rangle = \frac{2\langle G \rangle}{\pi e} \int_0^\infty d\epsilon \, \frac{\Delta^2 \sin \phi}{\epsilon^2 + \Delta^2} K\left( \frac{\Delta \sin(\phi/2)}{\sqrt{\epsilon^2 + \Delta^2}} \right). \tag{27}$$

Here $K(k)$ is the complete elliptic integral of the first kind with the modulus $k$. It arises here as the limit of the Legendre function with the degree $-1/2$.

Calculation of the energy integral yields

$$\langle I(\phi) \rangle = \frac{2\langle G \rangle \Delta}{\pi e} K^2\left( \sin \frac{\phi}{4} \right) \sin \phi. \tag{28}$$

We have thus established a rather unexpected fact: even in a strongly localized wire proportionality between the critical current and the normal state conductance (Ambegaokar-Baratoff relation) still holds on the level of average quantities. This may suggest a possible explanation of the experimental findings in Ref. [10]; more discussion of this topic in Sec. 6.

The result (28) was derived neglecting the potential term in the evolution equation (21). Accurately taking this term into account, even as a small perturbation, is quite a challenging mathematical task since the potential in Eq. (21) is not rotationally invariant in terms of the matrix $\tilde{Q}$. However, since only the lowest part of the Hamiltonian spectrum contributes to the supercurrent, it turns out that the effect of the potential term can be reduced to an energy-dependent quantization of the possible values of $q$:

$$q = \frac{\pi n}{\ln(\Delta_\xi/\epsilon)}, \qquad n = 1, 2, 3 \dots \tag{29}$$

As long as these discrete values are limited by the condition $q \ll 1$, we can simply replace the $q$ integral in Eq. (23) by the sum over $n$. Detailed derivation of this result is provided in Appendix A.

Summation over quantized $q$ modifies the integral (27) for the average supercurrent:

$$\langle I(\phi) \rangle = \frac{2\langle G \rangle}{\pi e} \int_0^\infty d\epsilon \, \frac{\Delta^2 \sin \phi}{\epsilon^2 + \Delta^2} K\left( \frac{\Delta \sin(\phi/2)}{\sqrt{\epsilon^2 + \Delta^2}} \right) F\left( \frac{\pi^2 L}{\xi \ln^2(\Delta_\xi/\epsilon)} \right). \tag{30}$$

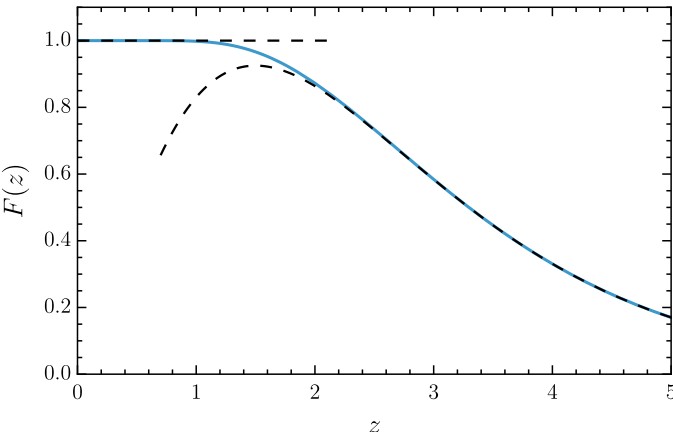

Figure 3: Crossover function $F(z)$ defined in Eq. (31).

In this more precise expression we have introduced the notation

$$F(z) = \frac{4z^{3/2}}{\sqrt{\pi}} \sum_{n=1}^{\infty} n^2 e^{-zn^2} \approx \begin{cases} 1, & z \ll 1, \\ \frac{4z^{3/2}}{\sqrt{\pi}} e^{-z}, & z \gg 1. \end{cases} \tag{31}$$

This function is shown in Fig. 3; it encapsulates all the necessary information about summation over discrete $q$. We have normalized it in such a way that in the limit $z \ll 1$, when the sum can be replaced by the integral, $F = 1$ and the previous result (28) is restored.

We thus observe that Eq. (28) remains valid provided $\xi \ll L \ll (\xi/\pi^2)\ln^2(\Delta_\xi/\Delta)$. For longer junctions we can estimate the current (30) taking advantage of the fact that the argument of $F$ slowly changes with $\epsilon$. We introduce a new integration variable $y = \ln(\Delta/\epsilon)$ and rewrite the integral (30) in the form

$$\langle I(\phi) \rangle = \frac{\langle G \rangle \Delta}{\pi e} \sin \phi \int_{-\infty}^{\infty} \frac{dy}{\cosh y} K\left( \frac{\sin(\phi/2)}{\sqrt{e^{-2y} + 1}} \right) F\left( \frac{\pi^2 L}{\xi[\ln(\Delta_\xi/\Delta) + y]^2} \right). \tag{32}$$

The first factor in the integrand decays exponentially for $|y| \gtrsim 1$. Hence we can neglect $y$ in the argument of $F$ in comparison to the large parameter $\ln(\Delta_\xi/\Delta)$. This allows us to replace the function $F$ by a constant, take it out of the integral, and then go back to the original integration over $\epsilon$ as in Eq. (27). The resulting expression for the average supercurrent becomes

$$\langle I(\phi) \rangle = \frac{2\langle G \rangle \Delta}{\pi e} F\left( \frac{\pi^2 L}{\xi \ln^2(\Delta_\xi/\Delta)} \right) K^2\left( \sin \frac{\phi}{4} \right) \sin \phi. \tag{33}$$

This generalizes Eq. (28) to junctions whose length exceeds $(\xi/\pi^2)\ln^2(\Delta_\xi/\Delta)$. For such long junctions we can also take the large $z$ asymptotics from Eq. (31). Using the average conductance from Eq. (24), we have for the supercurrent

$$\langle I(\phi) \rangle = \frac{2\pi^4 e \Delta}{\hbar \ln^3(\Delta_\xi/\Delta)} \exp\left( -\frac{\pi^2 L}{\xi \ln^2(\Delta_\xi/\Delta)} - \frac{L}{4\xi} \right) K^2\left( \sin \frac{\phi}{4} \right) \sin \phi. \tag{34}$$

Effectively, only the lowest eigenstate with $n = 1$ contributes to this result. We also observe that the Ambegaokar-Baratoff relation breaks down for such long junctions: while both the average conductance and the average supercurrent decay exponentially with $L/\xi$ the supercurrent decays faster than the conductance.

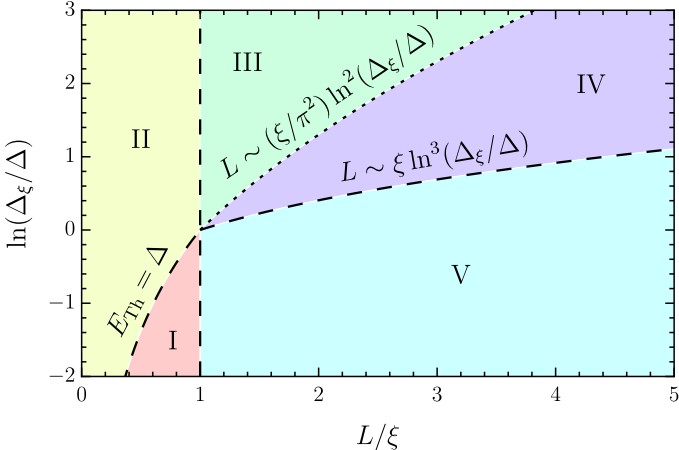

Figure 4: Phase diagram of various parametric regimes for the Josephson current depending on the ratio $L/\xi$ and $\ln(\Delta_\xi/\Delta)$. For relatively short junctions $L \ll \xi$, localization effects are not important and the two classical regimes are realized: regime I of classical long junction, see Eq. (12), and regime II of classical short junction, Eq. (11). In the opposite localized limit $L \gg \xi$, three cases arise. Regime III is described by Eq. (28). With further increasing the length $L$, it gradually crosses over to the regime IV with the supercurrent (34). The cases III and IV have the same current-phase relation and can be both described by a single expression (33). Finally, extremely long junctions fall into regime V with the supercurrent given by Eq. (36). Current-phase relations for all regimes are summarized in Fig. 5. Phase boundaries on this diagram are crossovers; expressions for the Josephson current are valid asymptotically deep inside the corresponding parameter range.

We summarize conditions for different limiting forms of the Josephson current in the parametric phase diagram Fig. 4. The limits of classical long and short junctions considered in Sec. 2 correspond to regimes I and II on this diagram, respectively. The localized limit of Eq. (33) covers both region III and IV while Eqs. (28) and (34) correspond to limits III and IV individually. The current-phase relations for different parametric regimes are shown in Fig. 5. It remains to consider the limit of very long junctions corresponding to the region V of the phase diagram.

For very long junctions the approximation used to derive Eq. (33) breaks down. This happens because neglecting $y$ in the argument of $F$ in Eq. (32) is not justified for very large $L$. Namely, when $L \gg \xi \ln^3(\Delta_\xi/\Delta)$, linear in $y$ correction to the argument of $F$ in Eq. (32), although relatively small, gets larger than 1. For the exponentially decaying function $F$ this leads to a significant change. We thus conclude that the result (33) and hence (34) is valid only for junctions whose length is limited from above by the condition $L \ll \xi \ln^3(\Delta_\xi/\Delta)$.

For extremely long junctions $L \gg \xi \ln^3(\Delta_\xi/\Delta)$ an alternative estimate for the integral (30) is needed. It turns out that main contribution to the integral comes from $\epsilon \ll \Delta$ in this case. We can thus set $\epsilon = 0$ in the first two factors in the integrand and retain integration only for the function $F$ replaced by its large $z$ asymptotics. We also introduce a new integration variable $z$ equal to the argument of $F$ and rewrite the integral as

$$\langle I(\phi) \rangle \sim \frac{eE_{\text{Th}}}{\hbar} \frac{L}{\xi} e^{-L/4\xi} \int_0^\infty dz \exp\left(-z - \pi\sqrt{\frac{L}{\xi z}}\right) K\left(\sin\frac{\phi}{2}\right) \sin\phi. \qquad (35)$$

Here we have explicitly substituted the average conductance from Eq. (24). The integral over $z$ can be computed by the standard saddle point approximation using the large parameter

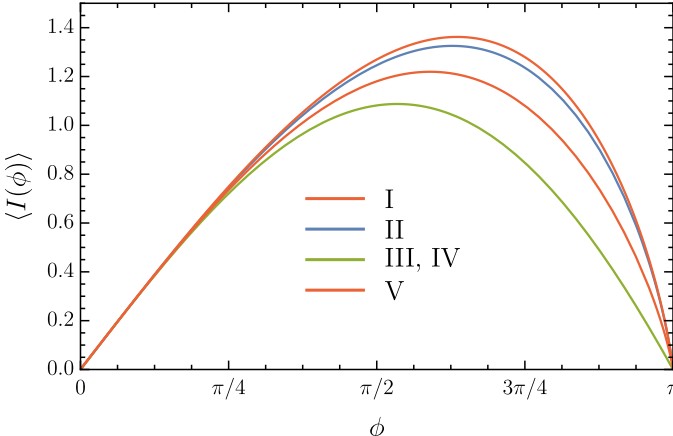

Figure 5: Current-phase relations in different regimes outlined in Fig. 4. The first two curves are for the classical ($L \ll \xi$) SNS junction. They coincide with the functions shown in Fig. 2. In the regimes III and IV, the phase dependence of the current is given by Eq. (33). In the regime V, the current is given by Eq. (36). All functions are normalized such that $\partial \langle I \rangle / \partial \phi = 1$ at $\phi = 0$.

$L/\xi \gg 1$. The saddle point lies at $z_* = (\pi^2 L/4\xi)^{1/3}$. The corresponding value of energy is indeed small $\epsilon_* \propto \Delta_\xi e^{-2z_*} \ll \Delta$ provided $L \gg \xi \ln^3(\Delta_\xi/\Delta)$. The resulting estimate for the average supercurrent is

$$\langle I(\phi) \rangle \sim \frac{eE_{\text{Th}}}{\hbar} \left( \frac{L}{\xi} \right)^{7/6} \exp\left[ -3 \left( \frac{\pi^2 L}{4\xi} \right)^{1/3} - \frac{L}{4\xi} \right] K\left( \sin \frac{\phi}{2} \right) \sin \phi. \tag{36}$$

Average supercurrent still decays faster than the average conductance but the additional decay factor is only a stretched exponential. Another remarkable feature of the result (36) is that this expression does not contain $\Delta$ explicitly.

We have so far considered all possible limiting forms of the supercurrent in the junction with $L \gg \xi$ and $\Delta_\xi \gg \Delta$. With increasing $L$ the supercurrent is given consecutively by Eqs. (28), (34), and (36). In other words, we cross over from region III to IV and then to V in the phase diagram Fig. 4 with increasing $L$ provided $\Delta_\xi \gg \Delta$. It remains to consider the case $\Delta_\xi \ll \Delta$.

Let us fix a large length $L \gg \xi$ and gradually decrease the ratio $\Delta_\xi/\Delta$ starting with some large value. This corresponds to moving down in the phase diagram Fig. 4 with some fixed ratio $L/\xi \gg 1$. We will eventually get to the parameter range V where Eq. (36) applies. There the average supercurrent is determined by the lowest eigenfunction of the Hamiltonian (21). Moreover, the energy integral (35) is dominated by the vicinity of the saddle point that corresponds to a very low energy $\epsilon_* \propto \Delta_\xi \exp[-2(\pi^2 L/4\xi)^{1/3}] \ll \Delta_\xi$. For such low energies the potential term in the Hamiltonian can still be treated perturbatively as described in Appendix A. We thus conclude that the result (36) remains valid also in the limit $\Delta_\xi \ll \Delta$ for a sufficiently long junction with $L \gg \xi$. Hence the region V of the phase diagram in Fig. 4 extends downwards below the line $\Delta_\xi = \Delta$. If we now fix the ratio $\Delta_\xi/\Delta \ll 1$ and gradually decrease $L$ starting from $L \gg \xi$, the expression (36) will smoothly transition into the classical long junction result (12) obtained from the Usadel equation. Indeed, both Eqs. (12) and (36) do not contain $\Delta$ explicitly and give $\langle I \rangle \sim eE_{\text{Th}}/\hbar$ when $L \sim \xi$ and $\langle G \rangle \sim e^2/\hbar$. We thus conclude that there is a direct crossover between regions I and V of the phase diagram in Fig. 4 at the line $L \sim \xi$.

# 6  Conclusion

In the present paper we have studied the interplay of the Josephson effect and Anderson localization in long strongly disordered SNS junctions. Different parametric regimes of the supercurrent are summarized in Fig. 4. When the length of the junction is shorter than the localization length, $L \ll \xi$, classical results derived from the Usadel equation apply. They correspond to the regions I and II on the phase diagram with the current given by Eqs. (12) and (11), respectively. For long junctions $L \gg \xi$, we identify another three qualitatively different asymptotic regimes. They correspond to regions III, IV, and V of the phase diagram in Fig. 4 and the average current is given by Eqs. (28), (34), and (36), respectively. The regions III and IV have the same functional dependence of the average supercurrent on the phase difference and can be covered by a single expression (33) with the function $F$ defined in Eq. (31). Current-phase relations for all asymptotic regimes are shown in Fig. 5.

It should be emphasized that in all strongly localized regimes with $L \gg \xi$, Josephson current has a wide distribution showing strong sample-to-sample fluctuations. It is therefore not appropriate to directly compare measurements on individual strongly localized SNS samples with the average results found in this paper. Instead, a measuring device should ideally include a large number of similar SNS junctions connected in parallel in order to facilitate averaging of the observed current. Theoretical study of higher moments of the supercurrent will shed more light on the distribution function of this strongly fluctuating quantity. This will be the subject of a separate publication.

One striking feature of our result is that the Ambegaokar-Baratoff relation between critical current and normal-state conductance does hold not only in the limit of classical short junction but also for much longer and strongly localized junctions. Namely, it is valid for regions II and III of the phase diagram Fig. 4 if one understands it as a relation between average current and average conductance. This to some extent may explain the result of the experiment of Frydman and Ovadyahu [10]. Indeed, in that experiment it was observed that for the junctions with length much larger than $\xi$ the Ambegaokar-Baratoff relation remains valid. The junctions measured in Ref. [10] were very wide and hence are not directly described by our one-dimensional theory. At the same time, large width of the junctions does to some extent facilitate averaging of mesoscopic fluctuations of the current. Hence, without direct comparison, we can claim that observation of the Ambegaokar-Baratoff relation in the strongly localized limit does qualitatively conform with our theory.

In all our calculations, we have assumed that the SNS junction contains only potential disorder and preserves both time-reversal and spin symmetries. This corresponds to the nonlinear sigma model of the orthogonal symmetry class. An alternative model might include strong spin-orbit interaction in the normal part of the junction, which would break the spin symmetry and result in the symplectic class instead. While this is unimportant for classical results in the limit $L \ll \xi$, localization phenomena are different in these two cases and the results for strongly localized junctions may differ. We have checked that Eq. (28) remains valid in the symplectic class while the expression (24) for the average conductance has a slightly different pre-exponential factor, see Ref. [19]. We thus conclude that the overall classification of different localized regimes in Fig. 4 remains valid also for junctions with strong spin-orbit interaction while some insignificant numerical factors may differ.

## Acknowledgements

We are grateful to M. Feigel'man, Ya. Fominov, N. Kishmar, and M. Skvortsov for fruitful and stimulating discussions.

## A  Transfer-matrix Hamiltonian

In this Appendix, we discuss low-lying eigenfunctions of the Hamiltonian (21) relevant for the calculation of the average supercurrent in the localized regime $L \gg \xi$. Kinetic term of the Hamiltonian has the form of the Laplace-Beltrami operator on the sigma-model manifold. Eigenfunctions of this operator can be constructed with the help of Iwasawa decomposition as was proposed in Refs. [17, 18] and carried out explicitly in Ref. [19].

Laplace-Beltrami operator separates into "radial" and "angular" parts when the parametrization (16) is used:

$$\nabla_Q^2 = \nabla_\theta^2 + \nabla_U^2. \tag{A.1}$$

We are mostly interested in the eigenfunctions of the radial part—zonal spherical functions—as was discussed in the main part of the paper. The radial operator acts only on the $\theta$ angles and has the form

$$\nabla_\theta^2 = \frac{1}{J}\left[ \frac{\partial}{\partial\theta_{\mathrm{B}}} J \frac{\partial}{\partial\theta_{\mathrm{B}}} + \frac{\partial}{\partial\theta_{\mathrm{B2}}} J \frac{\partial}{\partial\theta_{\mathrm{B2}}} + \frac{\partial}{\partial\theta_{\mathrm{F}}} J \frac{\partial}{\partial\theta_{\mathrm{F}}} \right]. \tag{A.2}$$

The Jacobian factor here is

$$J = \frac{\sinh\theta_{\mathrm{B}} \sinh\theta_{\mathrm{B2}} \sin^3\theta_{\mathrm{F}}}{[\cosh(\theta_{\mathrm{B}} + \theta_{\mathrm{B2}}) - \cos\theta_{\mathrm{F}}]^2 [\cosh(\theta_{\mathrm{B}} - \theta_{\mathrm{B2}}) - \cos\theta_{\mathrm{F}}]^2}. \tag{A.3}$$

One special eigenfunction of the Laplace-Beltrami operator is 1 ($\theta$-independent constant); the corresponding eigenvalue is 0. Other, nonzero eigenfunctions are enumerated by one discrete number $l = 0, 1, 2, \ldots$ and two positive continuous numbers $q_{1,2}$. The eigenvalues are

$$\nabla_\theta^2 L_{l,q_1,q_2} = -\left[ l(l+1) + q_1^2 + q_2^2 + 1/2 \right] L_{l,q_1,q_2}. \tag{A.4}$$

The lowest branch of the spectrum corresponds to $l = 0$. This set of eigenfunctions obeys an additional constraint $q_1 = q_2$. We will label them with a single index $q$. According to Ref. [19], these eigenfunctions are

$$L_q = (q^2 + 1/4)(\cos\theta_{\mathrm{F}} - \cosh\theta_{\mathrm{B}} \cosh\theta_{\mathrm{B2}}) P_{-1/2+iq}(\cosh\theta_{\mathrm{B}}) P_{-1/2+iq}(\cosh\theta_{\mathrm{B2}})$$
$$+ \sinh\theta_{\mathrm{B}} \sinh\theta_{\mathrm{B2}} P_{-1/2+iq}^1(\cosh\theta_{\mathrm{B}}) P_{-1/2+iq}^1(\cosh\theta_{\mathrm{B2}}) \quad \text{(A.5)}$$

and their eiganvalues are $2q^2 + 1/2$.

Zonal spherical functions represent a complete function basis in the space of angles $\theta$. We can use them to expand the delta function on the sigma model manifold [18, 19]:

$$\delta(Q - \Lambda) = 1 + \frac{\pi}{2} \int_0^\infty dq \frac{\tanh^2(\pi q)}{q^2 + 1/4} L_q(Q) + \ldots \tag{A.6}$$

Here we have omitted the contribution of all higher eigenfunctions with $l > 0$.

The delta function in Eq. (A.6) is the initial condition for the evolution equation (21). We can construct the solution to this equation for any value of $L$ by inserting an exponential factor with the corresponding eigenvalue into the integrand in Eq. (A.6). Setting $\theta_{\mathrm{B2}} = 0$ and using Eq. (A.5) we immediately arrive at the result (23).

In the calculation of the supercurrent, we have the evolution equation (21) with a different initial condition. As it is explained in the main text of the paper, we can apply a global rotation (25) and bring the initial point to $\Lambda$ in terms of the new matrix $\tilde{Q}$. Since the Laplace-Beltrami operator is invariant under any global rotations, the corresponding eigenfunctions retain their form (A.5) in terms of the new angles $\tilde{\theta}$, defined by Eqs. (26). However, the potential term in the Hamiltonian (21) becomes quite complicated in the new variables: it depends not only

on the $\tilde{\theta}$ angles but also on $\tilde{U}$. To overcome this complication, we will take advantage of the condition $\epsilon \ll \Delta_\xi$, which means that the potential term is important only at large values of $\tilde{\theta}$.

The radial Laplace-Beltrami operator retains its form (A.2) in terms of $\tilde{\theta}$. In the limit when both noncompact angles $\tilde{\theta}_B$ and $\tilde{\theta}_{B2}$ are large, this operator further splits into the part acting on $\tilde{\theta}_+ = \tilde{\theta}_B + \tilde{\theta}_{B2}$ and the part acting on $\tilde{\theta}_- = \tilde{\theta}_B - \tilde{\theta}_{B2}$ and $\tilde{\theta}_F$:

$$\nabla_{\tilde{\theta}}^2 \approx 2\left(\frac{\partial^2}{\partial \tilde{\theta}_+^2} - \frac{\partial}{\partial \tilde{\theta}_+}\right) + \frac{2}{J'}\frac{\partial}{\partial \tilde{\theta}_-}J'\frac{\partial}{\partial \tilde{\theta}_-} + \frac{1}{J'}\frac{\partial}{\partial \tilde{\theta}_F}J'\frac{\partial}{\partial \tilde{\theta}_F}, \qquad J' = \frac{e^{-\tilde{\theta}_+}\sin^3 \tilde{\theta}_F}{(\cosh \tilde{\theta}_- - \cos \tilde{\theta}_F)^2}. \quad \text{(A.7)}$$

The eigenfunction (A.5) depends only on $\tilde{\theta}_+$ in this limit. If we additionally assume $q \ll 1$, we can approximate $L_q(\tilde{\theta})$ up to a constant factor with the plane wave

$$L_q(\tilde{\theta}) \sim e^{\tilde{\theta}_+/2}\sin(q\tilde{\theta}_+). \quad \text{(A.8)}$$

It is easy to see that this plane wave is indeed an eigenfunction of the operator (A.7).

Let us now rewrite the eigenfunction $L_q(\tilde{\theta})$ in the original coordinates $\theta$ and $U$. In general, this will produce a very complicated function that depends on all the variables. However, for large $\tilde{\theta}_+$, we have an approximate identity

$$e^{\tilde{\theta}_+} \approx \text{str}(\Lambda \tilde{Q}) = e^{\theta_+}M(U), \qquad M(U) = \frac{1}{2}\text{tr}\left[Ue^{i\theta_S\sigma_y\tau_z}U^{-1}(\sigma_y\tau_z + \sigma_z\tau_y)\right]_{BB}. \quad \text{(A.9)}$$

With this value for $\tilde{\theta}_+$ and in the limit $q \ll 1$, the variables in the asymptotic plane wave function (A.8) decouple:

$$L_q(\tilde{\theta}) \approx L_q(\theta)\sqrt{M(U)}. \quad \text{(A.10)}$$

Here the radial part $L_q(\theta)$ is again given by Eq. (A.8) in terms of the original angle $\theta_+$. Since the Laplace-Beltrami operator retains its form under any rotations of $Q$, we conclude that the angular part of the factorized eigenfunction (A.10) obeys

$$\nabla_U^2\sqrt{M(U)} = 0. \quad \text{(A.11)}$$

This identity can be also verified directly by using an explicit parametrization of $U$ and taking the limit $\theta_+ \gg 1$ in the operator $\nabla_U^2$.

Now everything is prepared to include the potential term of the Hamiltonian (21) into our analysis. The potential term has much simpler form in the original coordinates: it depends only on $\theta$ but not on $U$. Moreover, it is relevant only for very large $\theta_+ \gtrsim \ln(\Delta_\xi/\epsilon)$. We can use the factorization (A.10) of the eigenfunction in the limit $\theta_+ \gg 1$ and modify the radial part $L_q(\theta)$ such that it obeys

$$\left[-\frac{\partial^2}{\partial \theta_+^2} + \frac{\partial}{\partial \theta_+} + \frac{\epsilon}{2\Delta_\xi}e^{\theta_+}\right]L_q(\theta) = (q^2 + 1/4)L_q(\theta). \quad \text{(A.12)}$$

This equation has the solution (up to a constant factor)

$$L_q(\theta) \sim e^{\theta_+/2}K_{2iq}\left(\sqrt{\frac{2\epsilon}{\Delta_\xi}}e^{\theta_+/2}\right). \quad \text{(A.13)}$$

In the limit $\theta_+ \ll \ln(\Delta_\xi/\epsilon)$ and $q \ll 1$, it simplifies to

$$L_q(\theta) \sim e^{\theta_+/2}\sin\left[q\left(\theta_+ - \ln\frac{\Delta_\xi}{\epsilon}\right)\right]. \quad \text{(A.14)}$$

We thus conclude that in the parametrically large interval $1 \ll \theta_+ \ll \ln(\Delta_\xi/\epsilon)$ the two asymptotics (A.8) and (A.14) of the eigenfunction $L_q(\theta)$ should be consistent with each other. This condition implies the values of $q$ are quantized as in Eq. (29).

At not very large values of the angles $\tilde{\theta}$, potential term in the Hamiltonian can be neglected and true eigenfunctions are still given by $L_q(\tilde{\theta})$ of Eq. (A.5). However, allowed values of $q$ should be quantized such that these eigenfunctions match with Eq. (A.13) at large values of the original angle $\theta_+$. Hence the evolution operator with the initial condition $\tilde{Q} = \Lambda$ in the presence of the potential term can be represented by Eq. (23) directly in terms of the angles $\tilde{\theta}$ but up to a replacement of the $q$ integral with the equivalent sum over discrete values of $q$.

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
