# Peer review of "Josephson effect in strongly disordered metallic wires"

_SciPost Physics_

## Round 1 · Referee Report · Alex Kamenev (Referee 1) · 2025-4-28

Report

The paper is devoted to calculations of the average Josephson current in an SNS system with the length of the normal region, N, exceeding its localization length. To this end the authors adopted the transfer-matrix treatment of the corresponding supersymmetric nonlinear sigma-model, pioneered by K. Efetov and M. Zirnbauer et al, and further refined by one of the authors with E. Khalaf. They found that the validity of the Ambegaokar-Baratoff (AB) relation, stating proportionality of the Josephson current and conductance, extends to a portion of the phase diagram covering a part of the localized regime. They also found two regimes, where the AB relation does not hold. I found these results novel, interesting, and deserving publication in the SciPost. I agree with the authors that the really interesting question is finding the entire probability distribution function of Josephson currents. Since the latter is a thermodynamic property, this task may prove to be substantially simpler than the distribution of conductances. It is conceivable that one can skip calculations of higher moments and directly evaluate a corresponding generating function.

Recommendation

Publish (easily meets expectations and criteria for this Journal; among top 50%)

  • validity: top
  • significance: good
  • originality: high
  • clarity: high
  • formatting: excellent
  • grammar: excellent

Author:  Pavel Ostrovsky  on 2025-06-16  [id 5574]

(in reply to Report 1 by Alex Kamenev on 2025-04-28)

We are extremely grateful to Prof. Kamenev for his very positive report.

---

## Round 1 · Referee Report · Anonymous (Referee 2) · 2025-4-30

Strengths

  1. Top-level application of the non-linear sigma model to a nontrivial problem;
  2. Clarity and pedagogical character of the exposition of derivations

Weaknesses

  1. No discussion of the possible role of various factors beyond the model considered (finite temperature, dephasing, etc), which are inevitable in realistic experimental settings

Report

This is top-level work addressing the effect of Anderson localization on the Josephson current in a disordered SNS junction, where the normal part is a multichannel disordered wire. By applying the non-linear sigma-model formalism, the authors determine the "phase diagram" with a multitude of localization-induced regimes. I strongly recommend the publication of the manuscript with minor modifications in response to the comments and questions described below.

Requested changes

  1. The actions (2) and (13) are written in the diffusive approximation, where the value of Delta is assumed to be exactly zero everywhere in the normal part of the junction. Of course, even in this case, the superconducting proximity effect is obtained from the sigma-model. Would the action be of the same type at the "more ballistic" level of the Eilenberger equation, or should one introduce some smooth profile of Delta(x) decaying into the bulk from the SN interface and describing the proximity effect in the effective action? In the present consideration, where Delta jumps at the interface, this might become important for establishing the boundary conditions for the Q field of the diffusive sigma-model. Did the authors consider the stability of their solution with respect to the variation of the boundary conditions? In particular, the authors mention that they neglect the gradient term when imposing their boundary conditions.
  2. The authors refer to Ref. [14] below Eq. (11), where they compare their result for short junctions with the formula derived within the scattering theory. Based on their results obtained for arbitrary lengths of the junctions, can the authors speculate how Eq. (35) of Ref. [14], expressing the Josephson current in terms of the eigenvalues of the transmission matrix, could be generalized to cover all the limiting cases without calculating full determinants, if possible? Given a direct relation between the sigma-model consideration and the transfer-matrix framework in quasi-1D systems, it can be expected that a rather compact general formula of the type of Eq. (35) of Ref. [14] exists.
  3. Equation (35) of Ref. [14] is derived at finite temperature. The inclusion of temperature makes the Josephson current a highly nonlinear function of the transmission eigenvalues. What is the effect of temperature on the results derived in the present manuscript (without considering dephasing)?
  4. Dephasing is inevitable in realistic systems. Therefore, in the discussion of the results, it would be useful to present simple speculations (without calculations) on the role of dephasing in the phase diagram obtained in the manuscript.

Recommendation

Publish (surpasses expectations and criteria for this Journal; among top 10%)

  • validity: top
  • significance: high
  • originality: high
  • clarity: top
  • formatting: perfect
  • grammar: excellent

Author:  Pavel Ostrovsky  on 2025-06-16  [id 5573]

(in reply to Report 2 on 2025-04-30)

We are grateful to the Referee for positive evaluation of our work and for constructive comments and criticism. Below we provide our response to the questions raised by the Referee and clarify changes made in the revised version of the manuscript.

1) We indeed have considered only the simplest boundary conditions with Delta abruptly dropping from its bulk superconducting value to zero at the interface. Delta is exactly zero in the normal part of the junction simply due to absence of Cooper attraction there. This fact does not rely on the type of electron dynamics (ballistic vs. diffusive) and holds true even in completely clean samples. A spatial variation (partial suppression) of Delta near the interface is possible only on the superconducting side and is known as the inverse proximity effect. This is usually a weak effect for the following reason. In the standard BCS theory Delta is determined by the self-consistency equation that involves a logarithmically divergent energy integral of the anomalous Green function. This integral collects contributions from a relatively wide range of energies and is cut off at the scale of the order of Debye energy. At the same time, proximity to the normal metal modifies the anomalous Green function only at much lower energies of order Delta itself. This means that the value of the integral and hence the value of Delta is only weakly sensitive to the proximity to normal metal. Moreover, in realistic SNS junctions, a narrow normal metal wire is usually attached to bulk and relatively rigid superconducting leads that further diminishes the inverse proximity effect.

Ballistic phenomena on the level of the Eilenberger equation are ruled out since we consider junctions whose length considerably exceeds the electron mean free path. Main focus of our study is on the localization physics that is relevant on the scale of the much longer localization length. Any possible remnants of ballistic effects are exponentially suppressed in our case.

2) Main assumption of the scattering matrix formalism of Ref. [14] is the energy independence of the scattering probability. It means that this approach can indeed be used to derive the supercurrent in relatively short junctions corresponding to regions II and III of our phase diagram in Fig. 4. In the regions I, IV and V, scattering probability essentially depends on energy and the general formula from Ref. [14] is not applicable. We have included two new paragraphs in the end of page 10 and on page 14 (the last paragraph before conclusion) discussing this topic.

3 and 4) Both finite temperature and dephasing can be treated in a similar fashion by limiting the range of energies in the spectral integrals (7) and (20). At a nonzero temperature, these integrals should be replaced by the sum over discrete fermionic Matsubara energies. Dephasing effectively excludes low energies (of order of the dephasing rate) from the integral. We have added discussion of this topic in two new paragraphs on page 15 in the conclusion section. Moreover, in a similar way the supercurrent can be suppressed by any mechanism that breaks time-reversal symmetry (e.g. magnetic impurities) even without dephasing. We have also mentioned this possibility in the new portion of our text.

---

## Round 2 · List of Changes

1. Two new paragraphs in the end of page 10 and on page 14 (the last paragraph before conclusion) discussing possible application of the scattering matrix formalism to the calculation of the supercurrent.

2. Two new paragraphs on page 15 in the conclusion section discussing effects of nonzero temperature, dephasing and time-reversal symmetry breaking.

3. New references [14, 17, 22, 23, 24]

---

## Editorial Decision

accepted_in_target_journal